# A Self-Assembly of Single Layer of Co Nanorods to Reveal the Magnetostatic Interaction Mechanism

**DOI:** 10.3390/nano12142499

**Published:** 2022-07-21

**Authors:** Hongyu Du, Min Zhang, Ke Yang, Baohe Li, Zhenhui Ma

**Affiliations:** Department of Physics, Beijing Technology and Business University, Beijing 100048, China; duhongyu2019@163.com (H.D.); tian123zmm@126.com (M.Z.); yangke_0205@163.com (K.Y.)

**Keywords:** nanomagnets, Co nanorods, solvothermal route, alcohol–thermal method, magnetic interaction

## Abstract

In this work, we report a self-assembly method to fabricate a single layer of Co nanorods to study their magnetostatic interaction behavior. The Co nanorods with cambered and flat tips were synthesized by using a solvothermal route and an alcohol–thermal method, respectively. Both of them represent hard magnetic features. Co nanorods with cambered tips have an average diameter of 10 nm and length of 100 nm with coercivity of 6.4 kOe, and flat-tip nanorods with a 30 nm diameter and 100 nm length exhibit a coercivity of 4.9 kOe. They are further assembled on the surface of water in assistance of surfactants. The results demonstrate that the assembly type is dependent on the magnetic induction lines direction. For Co nanorods with flat tips, most of magnetic induction lines are parallel to the length direction, leading to an assembly that is tip to tip. For Co nanorods with cambered tips, they are prone to holding together side by side for their random magnetic induction lines. Under an applied field, the Co nanorods with flat tips can be further aligned into a single layer of Co nanorods. Our work gives a possible mechanism for the magnetic interaction of Co nanorods and provides a method to study their magnetic behavior.

## 1. Introduction

Nanomagnets, with strong magnetic properties and small volume, have been considered to be the key materials to magnetic and electronic devices that exhibit important applications in artificial intelligence, intelligent robots, wind turbines, and electromobiles [1,2,3,4,5,6,7,8,9,10,11,12]. Compared with rare-earth-based nanomagnets, the rare-earth-free nanomagnets earn more attention for their high chemical stability and low cost [13,14,15,16]. However, these rare-earth-free nanomagnets present small coercivity due to their relatively low magnetocrystalline anisotropy. One strategy to fabricate one-dimensional nanowires or nanorods by the special methods can resolve above problem, since the direction along nanorods length has the lowest demagnetizing field [17,18,19]. As a result, the shape anisotropy can be summed to magnetocrystalline anisotropy, leading to these nanorods with a larger coercivity than spherical particles. When an external field is applied in these particles, they can orient along the external field direction. Therefore, the system of magnetic nanorods along the applied field will exhibit an enhanced coercivity, high remanence (*M**_r_*), and large energy product. 

Hexagonal structured Co has a magnetocrystalline anisotropy constant (K_1_ = 440 kJ/m^3^), and its bulk has a high saturation magnetization (M_s_ = 160 emu/g) [20]. Therefore, in the last few decades, Co was always considered to be a soft magnetic material since its magnetocrystalline anisotropy is far smaller than those of some well-known hard magnetic structures, such as samarium−cobalt (SmCo_5_) (17,200 kJ/m^3^) and neodymium iron boron (Nd_2_Fe_14_B) (4900 kJ/m^3^) [21,22,23,24]. Recently, the Co nanowires have been a potential candidate for rare-earth-free nanomagnets because the Co nanowires with their length direction along [002] (the easy magnetization axis direction) can obtain a large coercivity. To obtain such shape-anisotropic Co nanowires, a conventional approach via electrochemical deposition of Co into a porous alumina template was employed, which yielded an enhanced coercivity from an aligned Co nanowire array, but it is lower than the theoretical value because of the polycrystalline structure [25]. Recently, a solution-phase synthesis was used to reduce a cobalt salt using an organic polyalcohol. The resulted Co nanowires possessed a single crystal structure, and their lengths (50–300 nm) and diameters (5–30 nm) can be tuned to achieve high performance [26,27,28,29]. As a consequence, the aligned Co nanomagnet fabricated by nanowires exhibits an ultrahigh coercivity up to 10 kOe, which exceeded the theoretical value of bulk Co material (7.6 kOe) [27]. The high coercivity originated from a sum of magnetocrystalline anisotropy field (7.6 kOe) and shape anisotropy field (9 kOe) [27].

It is worth noting that the magnetic alignment is a key parameter to obtain large coercivity and high *M*_r_. To obtain an aligned nanomagnet, these as-prepared Co nanowires or nanorods have to be mixed with epoxy and aligned under an applied field [26,27]. In such a nanomagnet, the aligned Co nanowires were stacked layer by layer, and the magnetostatic interaction among the Co layers will have an obvious influence on alignment and magnetic performance. However, the single layer of aligned Co nanorods was very hard to prepare by using the current methods; however, it was valuable to study the magnetostatic interaction of nanomagnets.

Here, we developed a novel method to obtain a single layer of aligned Co nanorods. First, we synthesized different shaped Co nanorods by solvothermal route (the nanorods with flat tips) and alcohol–thermal method (the nanorods with cambered tips), respectively. These nanorods were further aligned by using a self-assembly strategy. As a result, a well-aligned single layer of nanorods was obtained under an applied field. Their self-assembly behaviors were further studied without an applied field. It was found that the nanorods with flat tips are prone to aligning tip to tip, and the nanorods with cambered tips are easy to hold together side by side. Furthermore, we analyzed the magnetic interaction mechanism of Co nanorods with different tips to demonstrate their assembly behaviors. Our work provides an important reference to study the magnetostatic interaction of shape anisotropic nanomagnets.

## 2. Experiments and Methods

### 2.1. Chemicals and Materials

All chemicals were used without further purifications. Cobalt chloride hexahydrate (CoCl_2_·6H_2_O, 98%, M_w_ = 237.93), sodium laurate (C_11_H_23_COONa, ≥99%, M_w_ = 222.30), oleylamine (OAm, 70%, M_w_ = 267.49), oleic acid (OA, 90%, M_w_ = 282.46), 2-butanediol (BEG, 98%, M_w_ = 90.12), Ruthenium chloride (III) hydrate (RuCl_3_·xH_2_O, 99.98%, M_w_ = 207.43), hexadecylamine (HDA, 98%, M_w_ = 241.46), hexane (≥99%, M_w_ = 86.18), methylbenzene (=99.8%, M_w_ = 82.14), and chloroform (≥99.8%, M_w_ = 119.38) were purchased from Sigma-Aldrich, St. Louis, MO, USA.

### 2.2. The Synthesis of Cobalt (II) Laurate

The Cobalt (II) laurate (Co (C_11_H_23_COO) _2_) was prepared by mixing C_11_H_23_COONa and CoCl_2_. In a typical reaction, 9.3366 g of C_11_H_23_COONa was dissolved in 30 mL of deionized water, with a mechanical stirrer, in three-neck flask, which was heated to 60 °C, using a water bath. Then 5.000 g of CoCl_2_ was dissolved in 10 mL deionized water in another flask, forming CoCl_2_ aqueous solution. The latter was added into former system, dropwise, with vigorous stirring. The mixture was stirred for a further 30 min and kept at 60 °C to obtain a purple precipitate, which was centrifuged at 8000 rpm for 8 min, and washed three times with 50 mL deionized water and methanol, respectively. The as-prepared precipitate was further dried in an air oven at 60 °C.

### 2.3. The Synthesis of Cobalt Nanorods Using a Solvothermal Route

The Co nanorods were first synthesized by reducing Co-laurate in BEG. In a typical synthesis, 2.0700 g of Co-laurate, 0.5810 g of HDA, and 0.0037 g of RuCl_3_ were dissolved in 60 mL of BEG in a three-neck flask, which was heated to 90 °C for 30 min to obtain a uniform solution. Then the solution was transferred into a Teflon enclosure (100 mL) and was further heated to 250 °C for 60 min, with a heating rate of 15 °C/min for a solvothermal route. After cooling to room temperature, the black magnetic product was precipitated and washed by toluene by centrifuging at 6000 rpm for 5 min at least 3 times.

### 2.4. The Synthesis of Cobalt Nanorods Using an Alcohol–Thermal Method

In a representative synthesis, 0.6113 g of Co-laurate, 0.0077 g of RuCl_3_, and 0.4640 g of NaOH were added to 15.5 mL of BEG. Under mechanical stirring, the reaction mixture was heated to 100 °C for 30 min to obtain the uniform solution. Then the solution was further heated to 175 °C, with a temperature ramping rate of 7 °C/min, and was maintained at this temperature for 30 min. After cooling to room temperature, the black powder was recovered by centrifugation at 6000 rpm for 5 min and washed with ethanol for 3 times. For short Co nanorods, the mechanical stirring rate was 100 r/min; and for the long Co nanorods, the rate was set to 50 r/min.

### 2.5. The Self-Assembly of Co Nanorods 

To obtain single-layer Co nanorods, we used OA and OAm, with a volume ratio of 1:3, as the surfactants. First, 0.0010 g of Co nanorods was dispersed 20 mL of above the mixed solution to achieve the ligand exchange under ultrasonic shock for 24 h. Then these nanorods were recovered by centrifugation and washed with hexane 3 times. Then the centrifuged Co nanorods were dispersed again in 20 mL methylbenzene, using ultrasonic shock. One or two droplets of Co nanorods/methylbenzene were dropped on the smooth surface of the water, and the system was sealed by using glass sheets to make the methylbenzene evaporate slowly for 4 h. A TEM grid was used to collect the self-assembly samples. For the magnetic field assisted self-assembly, a magnet was closed the water surface during the evaporation of organic solution. The other surfactants, HDA and PVP, and the other solutions, hexane and chloroform, were employed to replace the OA/OAm and methylbenzene, respectively. However, they failed to get the well-aligned Co layers. 

### 2.6. Characterization

The Co nanorods’ structure was studied by X-ray diffraction (XRD, D/MAX 2200 PC, Rigaku Corporation, Tokyo, Japan) with Cu-K_α_ radiation (λ = 0.15418 nm). The microstructure and morphology of the samples were analyzed by transmission electron microscopy (TEM, Philips CM 20, Philips, Amsterdam, Netherlands). HRTEM was performed on JEM-2100F (Japan Electronics jeol, Tokyo, Japan). The magnetic properties were measured at room temperature, using a vibrating sample magnetometer (VSM) under a maximum applied field of 30 kOe. Moreover, all of the magnetic properties were measured by using the random aligned Co rods.

## 3. Results and Discussions

### 3.1. The Self-Assembly of Co Nanorods with Cambered Tips

We first synthesized Co nanorods with cambered tips by solvothermal route by using a hydrothermal reactor, as shown in Figure 1. The XRD pattern (Figure 1A) demonstrates that the solvothermal route yields hexagonal-structured Co particles. All broad peaks have a good match with standard Co (JCPDS No. 01-1278) pattern. There are obvious differences among the width of the diffraction peaks (100), (002), and (101). The broader (002) plane indicates the smaller size perpendicular to [002] than other directions, suggesting the shape of the anisotropic particles that were obtained. The TEM results further confirm the successful synthesis of shape anisotropic Co nanoparticles. It can be seen from Figure 1B,C that the solvothermal reaction generates the uniform Co nanorods, with their diameter of about 10 nm and length of 100–150 nm. These as-prepared Co nanorods have obvious cambered tips and are prone to being distributed side by side. The microstructure of the Co nanorod was further observed by using HRTEM, as shown in Figure 1D. Along the length direction, the interplanar spacing of 0.20 nm was observed, matching well with the hcp-Co (002) plane, thus indicating that the length direction is consistent with the [002] direction. As we known, the [002] direction is the easy magnetization axis (c-axis). Therefore, such shape can contribute a large shape anisotropy, which can combine with magnetocrystalline anisotropy and give a high coercivity.

We tried to make these Co nanorods self-assemble by the molecular force of surfactant without any magnetic field. The OA and OAm were employed as the surfactant, and the results are shown in Figure 2. From the TEM images in Figure 2A–C, we can see that the Co nanorods were well aligned into an assembly. Unfortunately, these Co nanorods are not aligned along a particular direction. These self-assembly presents radial, and the direction is from the center to all round. This phenomenon can be explained by the fact that some aggregates of Co nanorods become a hard magnetic core, similar to a spherical magnet, which yields a strong radial magnetic field, making other nanorods align along the magnetic field (Figure 2B). We can further observe that these nanorods are aligned side by side (Figure 2C), which may be caused by the magnetostatic interactions among nanorods. We also tried to use other surfactants to achieve the self-assembly, but they failed to align well due to the strong magnetic interaction. This result demonstrates that the magnetic interactions among nanorods are far stronger than the molecular interactions of surfactants. We measured the magnetic hysteresis loop of random Co nanorods at room temperature (Figure 2D), which exhibits a large coercivity of 6.4 kOe and high M_s_ of 148.3 emu/g under a 2.5 kOe field. Once these nanorods are aligned into a nanomagnet, using a magnetic field, their coercivity will exceed 10 kOe, being an excellent candidate for strong magnetic materials [30].

### 3.2. The Self-Assembly of Co Nanorods with Flat Tips

As a comparison, we also synthesized Co nanorods with flat tips by using an alcohol–thermal method, using a reflux set, as shown in Figure 3. The XRD pattern (Figure 3A; the samples were prepared with a stirring rate of 100 r/min) can correspond to hexagonal structure Co very well. Moreover, we also can find the different peaks width for the (100), (002), and (101) plane, thus indicating that the as-prepared Co particles have a shape anisotropy. Different from Co nanorods obtained from solvothermal method, the diffraction peaks of Co (002) planes exhibit an obvious enhancement when compared with the standard peaks. This may suggest that these Co nanorods have a self-orientation effect. Figure 3B,C shows the TEM images of Co nanorods. The uniform Co nanorods display a bamboo-like shape with an average diameter of ~30 nm and length of ~100 nm. It is interesting that these Co nanorods have larger tips than their body, and their tips exhibit a flat plane. Without any magnetic field or surfactant, several Co nanorods are self-oriented tip to tip, forming a bamboo-like shape, which is totally different from these Co nanorods with cambered tips. We further synthesized a kind of longer Co nanorods by using a lower stirring rate of 50 r/min, as shown in Figure 3D. With the decrease in the stirring rate, there is no obvious change for the diameter of as-prepared Co nanorods, but their length does increase to 200 from 100 nm. Certainly, these longer Co nanorods still represent larger tips and smaller bodies, and they also have a self-orientation effect by going tip to tip. This demonstrates that the mechanical stirring can interrupt the growth of nanorods from the length direction; this can be explained by the fact that the growth of long Co nanorods requires Co^2+^ continuously feeding. With the increase of stirring rate, homogenizing the solution can hinder local increment of Co^2+^ concentration in the solution [31,32]. As a result, the growth of Co nanorods from the length direction was interrupted, forming multiple shorter nanorods.

The magnetic hysteresis loop of random Co nanorods with the length of 100 nm was measured at room temperature, using VSM. As shown in Figure 4, these Co nanorods exhibit a coercivity of 4.9 kOe and M_s_ of 146.5 emu/g, at an applied field of 2.5 kOe. Compared with Co nanorods with cambered tips, these Co nanorods with flat tips exhibit a lower magnetic performance. This may be ascribed to two reasons. On the one hand, the ratio of length to diameter (L/D) is a key factor to the coercivity due to the shape anisotropy. In general, the higher ratio will lead to larger coercivity [33]. In our work, the Co nanorods with cambered tips have an L/D of ~10, while the L/D value for Co nanorods with flat tips is only ~4. On the other hand, the Co nanorods with flat tips have smaller coercivity than cambered tips, even though they have the similar L/D. Because Co nanorods with flat tips have more defects (stacking fault) between the tips and bodies, this can lead to a high demagnetizing field, and it causes the low coercivity [34].

The coercive mechanism of Co nanorods is totally different from small-sized Co nanoparticles or clusters. As we know, the magnetic particles have the largest coercivity at the key size from the multiple-domain to single-domain structure, and the key size is about 30 nm for Co particles [17]. Thus, our Co nanorods with a diameter of about 10 nm and length of 100–150 nm have been display a relatively higher coercivity than small-sized Co particles (several nanometers). Furthermore, these Co nanorods have been confirmed to be single-domain structures in previous work [27]. Moreover, for Co clusters with a small size, they are superparamagnetic and, thus, have no coercivity at room temperature, since their magnetocrystalline anisotropy at this size cannot compete with thermal disturbance [35]. More important, the coercivity of hcp-Co nanorods is mainly originated from shape anisotropy rather than magnetocrystalline anisotropy. In other words, the high L/D leads to their large coercivity, which can be confirmed by many reports [33].

These as-prepared Co nanorods with flat tips were further assembled on the water solution surface by using OA and OAm as the surfactant. Without any external field, driven by the molecular force of surfactant and magnetostatic force, these Co nanorods tend to align tip to tip and form long nanochains, but there is obvious space between aligned nanochains (Figure 5A). Although this alignment cannot reach our expectation, most of as-formed Co nanochains do present consistent direction, which is an obvious improvement compared to the Co nanorods with cambered tips. We also can observe the well-aligned Co nanorods in some view of TEM, as shown in Figure 5B. In these regions, these nanorods first form straight nanochains, and then these nanochains assemble into a single layer. Unfortunately, these regions are isolated from each other, making it hard to form high quality single-layer self-assembly. This result may be caused by magnetostatic interaction, which is discussed in the next section in detail. To obtain high-quality Co nanorods assembly, an external field was applied during the self-assembly process. The results are shown in Figure 5C,D. According to the XRD pattern in Figure 5C, the assembled Co nanorods under external field only display a single diffraction peak around 44.5°, which can be indexed to hcp-Co (002) plane. As we know, the easy axis of Co nanorods is the [002] direction, which is also the length direction. When these nanorods were aligned, all of Co nanorods displayed the same direction. As a result, the X-ray only detected the (002) plane. The TEM image in Figure 5D demonstrates that these Co nanorods with ~100 nm length were well aligned into a single layer along the magnetic field direction, further confirming the XRD result. We further aligned Co nanorods with a ~200 nm length under the same conditions, but it failed to obtain good alignment; this can be explained by the fact that the stronger properties of Co nanorods make them unfavorable to align due to the strong interaction with each other.

### 3.3. The Discussion of Magnetostatic Interaction Mechanism

During the self-assembly process, there is a competition between molecular interaction force and magnetostatic force. The former drives these nanorods to align well, while the latter makes them random to reduce the magnetostatic energy. In our work, the magnetostatic force was obviously stronger than the molecular interaction force. Therefore, we should analyze their behavior from the view of magnetostatic interaction. 

We analyzed the magnetostatic interaction mechanism by using magnetic induction line distribution, as shown in Figure 6. All Co nanorods can be regarded as a tiny magnet at the nanoscale, and the two tips of nanorods correspond to their N–S poles (Figure 6(A1,B1)) [36]. Moreover, the magnetic induction lines at two tips should be perpendicular to their tips. 

For the Co nanorods with flat tips, they have larger tips. Therefore, most of magnetic induction lines are perpendicular to the flat tips and, thus, parallel to the [002] direction (Figure 6(A1)). As a result, these Co nanorods are prone to holding together tip to tip since the magnetic induction lines direction for the N pole of one Co nanorod is consistent with the S pole of another Co nanorod (Figure 6(A2)). If these nanochains assembled by Co nanorods had the same direction for N–S poles, they would present mutual exclusion, and, thus, these nanochains would be separated from each other (Figure 6(A3)). If they had the opposite N–S poles, they would hold together again by connecting the third nanochains and rotate at a certain angle to keep the minimum energy (Figure 6(A4,A5)). When an external field was applied, these nanochains can rotate into the magnetic field direction to achieve alignment.

For the Co nanorods with cambered tips, they have smaller tips. Due to the special morphology, they only have very few magnetic induction lines parallel to the c-axis. Most of the magnetic inductions line are random (Figure 6(B1)). Therefore, tip-to-tip assembly is very unstable. It needs to rotate at a certain angle to reduce the energy of system (Figure 6(B2)). On this occasion, these nanorods are prone to holding together side by side for their random magnetic induction lines, since there are some angles between most of magnetic induction lines and nanorods c-axis (Figure 6(B3,B4)). Despite that, the alignment side by side was confined to several nanorods [37]. These clusters assembled by several nanorods with the same direction can be regarded as a “magnetic domain”, and the system is similar to a “multidomain magnet” (Figure 6(B5)). To keep the minimum energy of the whole system, these “magnetic domain” directions are distributed randomly. When an external field or self-generated field (some nanorods aggregate similar to a magnet, as seen in Figure 2B) is applied, these “magnetic domains” can rotate into the magnetic field direction to achieve alignment.

## 4. Conclusions

In summary, we synthesized two kinds of Co nanorods, one with cambered tips and another with flat tips, and achieved a self-assembly of a single layer of Co nanorods to study their magnetostatic interaction behavior. The as-synthesized Co nanorods with cambered tips have an average diameter of 10 nm and length of 100 nm, with coercivity of 6.4 kOe; and flat-tip nanorods with a 30 nm diameter and 100 nm length exhibit a coercivity of 4.9 kOe. These Co nanorods were first assembled on the surface of water, without a magnetic field. It is found that the Co nanorods with cambered tips are prone to aligning side by side, while the Co nanorods with flat tips are easy to hold together tip to tip due to the magnetostatic interaction. Under an applied field, the Co nanorods with flat tips can be further aligned into a single layer of Co nanorods. Furthermore, we studied the magnetostatic interaction mechanism, using magnetic induction lines. Because each nanorod can be considered as a magnet, the flay tips of Co nanorods are similar to the N and S poles of a magnet. Therefore, they are easy to hold together tip to tip since most of magnetic induction lines are parallel to the length direction. For Co nanorods with cambered tips, they are prone to holding together side by side to form clusters for their random magnetic induction lines. Our work provides a method to study magnetic interaction of shape anisotropy.

## Figures and Tables

**Figure 1 nanomaterials-12-02499-f001:**
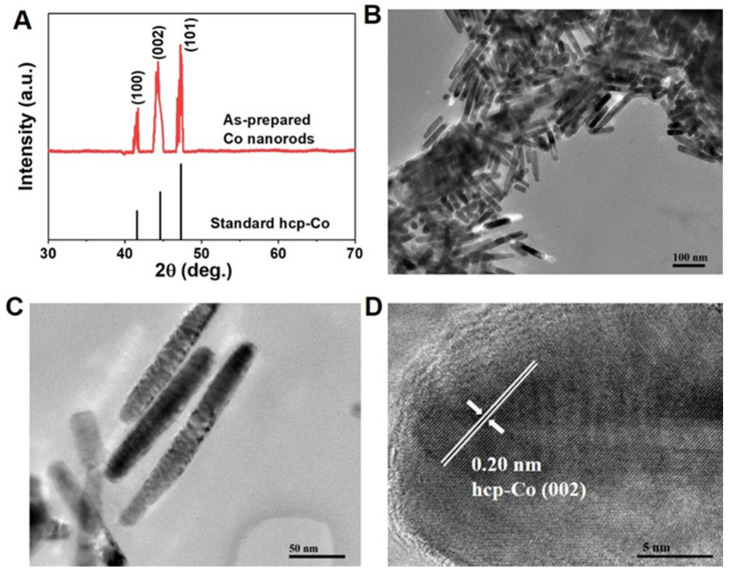
The Co nanorods with cambered tips by solvothermal route: (**A**) XRD pattern compared with standard hcp-Co (JCPDS No. 01-1278); (**B**,**C**) TEM images with different magnification; (**D**) HRTEM image.

**Figure 2 nanomaterials-12-02499-f002:**
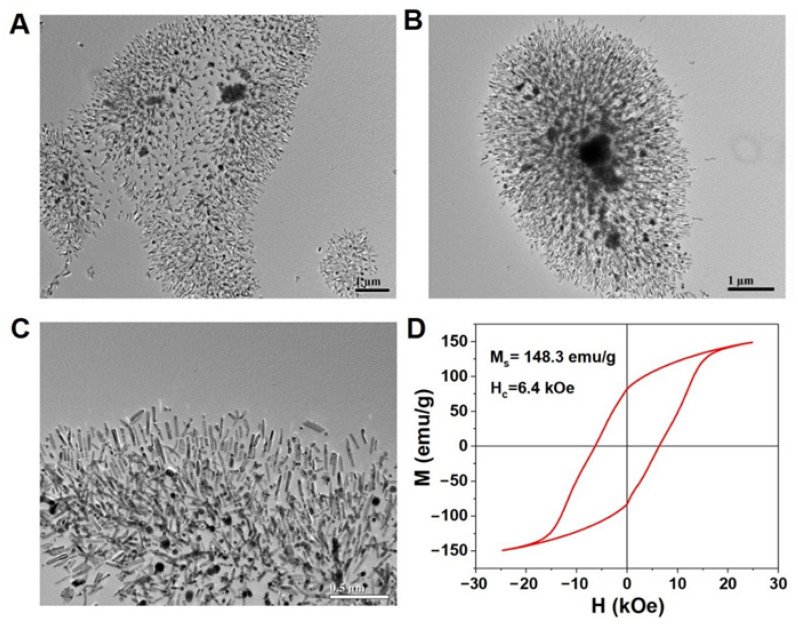
(**A**–**C**) The TEM images of self-assembly Co nanorods with cambered tips by solvothermal route; (**D**) Magnetic hysteresis loop of Co nanorods at room temperature.

**Figure 3 nanomaterials-12-02499-f003:**
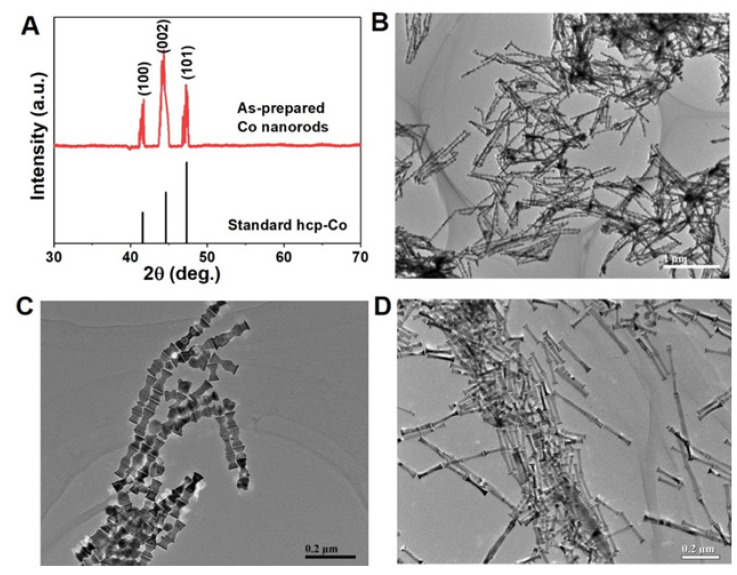
The Co nanorods with flat tips by alcohol–thermal method. (**A**) XRD pattern compared with standard hcp-Co (JCPDS No. 01-1278); (**B**,**C**) TEM images of Co nanorods with 100 r/min stirring rate; (**D**) TEM image of Co nanorods with 50 r/min stirring rate.

**Figure 4 nanomaterials-12-02499-f004:**
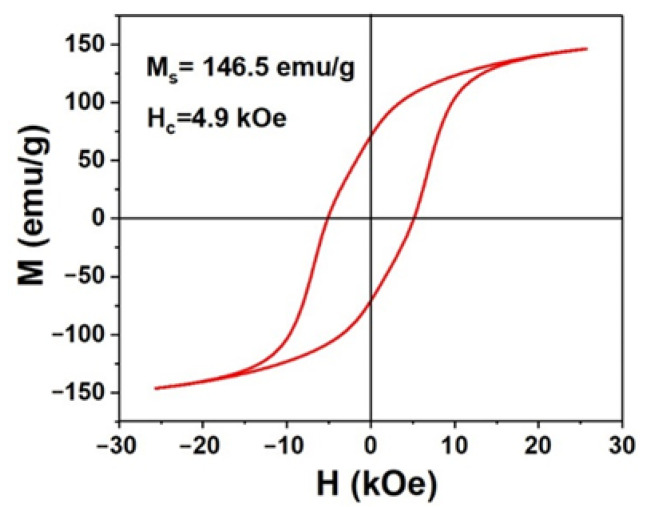
Magnetic hysteresis loop of Co nanorods with flat tips at room temperature.

**Figure 5 nanomaterials-12-02499-f005:**
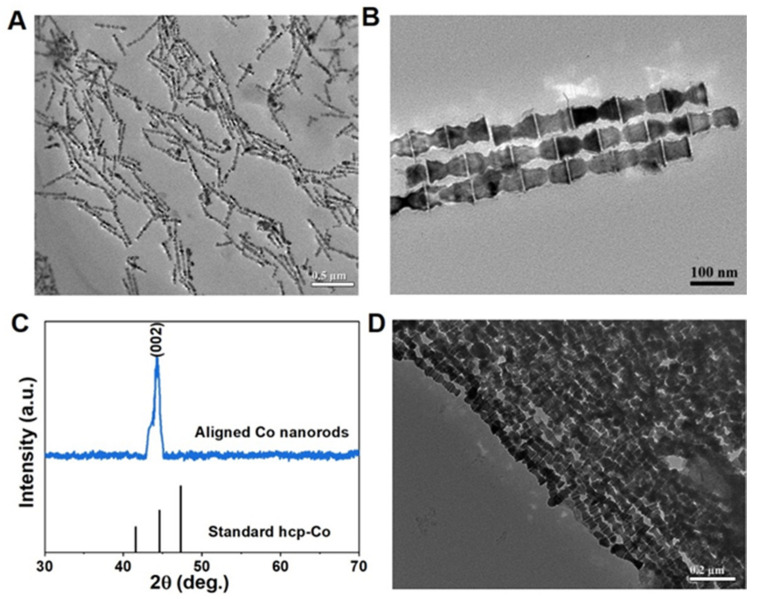
(**A**,**B**) The TEM images of self-assembly Co nanorods with flat tips, without applied field; (**C**) XRD pattern of aligned Co nanorods under magnetic field compared with standard hcp-Co (JCPDS No. 01-1278); (**D**) TEM image of aligned Co nanorods under magnetic field.

**Figure 6 nanomaterials-12-02499-f006:**
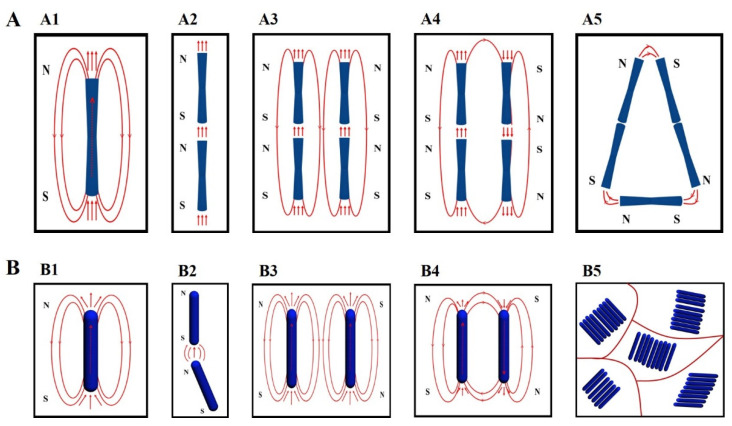
The scheme of magnetostatic interaction mechanism: (**A**) Co nanorods with flat tips and (**B**) Co nanorods with cambered tips.

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
