# Peer review of "A Self-Assembly of Single Layer of Co Nanorods to Reveal the Magnetostatic Interaction Mechanism"

_nanomaterials, 2022, doi:10.3390/nano12142499_

Round 1

Reviewer 1 Report

The paper by Du et al, reports the self-assembly of single layer of Co nanorods for the purpose of revealing the magnetostatic interaction mechanism. No optical discussion was conducted and there was no clear application of the reported materials. Although the synthesis steps were well conducted, and the structure of the manuscript was of scientific presentation, the authors would still need to revise the manuscript in the light of below concerns raised by this Reviewer

1) Some of the claimed results have no literature basis and were left very loosely used. For example, Authors claim that the mechanical stirring was able to interrupt the growth of nanorods from the length direction. In this, there was no citation or reference to previous reports or observations. Authors need to describe and show the mechanism of how mechanical stirring can interrupt the growth of nanorods from the length direction.

2) According to Figure 5C, the assembled Co nanorods under external field only displayed a single diffraction peak around 44.5o, which can be indexed to hcp-Co (002) plane. Authors should explain the reason behind just one diffraction peak.

83) Authors are advised to also carry out optical characterization of the Co nanoparticles in order to study the optical behavior and also infer any relationship between the morphology of the nanoparticles with their respective optical behavior

Author Response

Dear Reviewer,

Thank you for your e-mail and forwarding the reviewers’ comments on our manuscript entitled “A self-assembly of single layer of Co nanorods to reveal the magnetostatic interaction mechanism”. We thank the reviewers for their constructive comments and kind recommendation. We have revised the manuscript based on their comments. The revised sections are marked in red for your and reviewer’s references. The responses to all questions are given below:

1) Some of the claimed results have no literature basis and were left very loosely used. For example, Authors claim that the mechanical stirring was able to interrupt the growth of nanorods from the length direction. In this, there was no citation or reference to previous reports or observations. Authors need to describe and show the mechanism of how mechanical stirring can interrupt the growth of nanorods from the length direction.

Thank you for the good suggestions. We have added some literatures to demonstrate the mechanism of how mechanical stirring can interrupt the growth of nanorods from the length direction. Some sentences were added in the revised manuscript. (Chemical Engineering Journal 414 (2021) 128711; Journal of Experimental Nanoscience 10.18 (2015): 1387-1400).

The growth of long Co nanorods needs Co2+ continuously feeding. With the increase of stirring rate, homogenizing of solution can hinder local increment of Co2+ concentration in the solution. As a result, the growth of Co nanorods from the length direction was interrupted, forming multiple shorter nanorods.

2) According to Figure 5C, the assembled Co nanorods under external field only displayed a single diffraction peak around 44.5o, which can be indexed to hcp-Co (002) plane. Authors should explain the reason behind just one diffraction peak.

Thank you for raising this question. As we known, the easy axis of Co nanorods is [002] direction, which is also the length direction. When these nanorods were aligned, all of Co nanorods display the same direction. As a result, the X-ray only detected the (002) plane. These sentences are added in revision.

3) Authors are advised to also carry out optical characterization of the Co nanoparticles in order to study the optical behavior and also infer any relationship between the morphology of the nanoparticles with their respective optical behavior.

Thank you for your good suggestions. Co nanoparticles are a kind of typical magnetic materials. There are few reports on their optical behaviors. We expect that we can cooperate with you to study their optical properties.

We believe we have answered all questions raised by you and revised the paper accordingly. The new version now is ready to be accepted for publication. Thank you for your time and consideration.

Best regards,

Zhenhui Ma

Reviewer 2 Report

in this study, authors report a self-assembly method to fabricate single layer of Co nanorods to study their magnetostatic interaction behavior. For this purpose, solvothermal route was used. Also, the magnetostatic interaction mechanism was study by details. In my opinion, this work can be published in this journal after some corrections:

1-Mw and purity of reagent were missed in experimental section.

2- author should be explained the effects of nanorod shape on magnetic properties of products

3-it is important to use various surfactant agent in e self-assembly procedure

4- JCPD card no is necessary for all XRD patterns.

5- Quality of Fig. 6 should be improved

6- Recent advances on synthesis, mechanism process and physicho-chemical properties are advised included, e.g. A: doi: 10.1016/j.actamat.2020.01.044, B: doi: 10.1039/D2NR00488G, C: doi: 10.1002/aenm.201502588, D: doi: 10.3390/nano12060982, E: doi: 10.1016/j.measurement.2021.110527, F: doi: 10.1016/j.sna.2021.112789, G: doi: 10.1016/j.measurement.2020.108409, H: doi: 10.1109/TTE.2021.3104876 and I: https://doi.org/10.1002/adem.202101680

7- The novelty results should be highlighted in conclusion section.

Author Response

Dear Reviewer,

Thank you for your e-mail and forwarding the reviewers’ comments on our manuscript entitled “A self-assembly of single layer of Co nanorods to reveal the magnetostatic interaction mechanism”. We thank the reviewers for their constructive comments and kind recommendation. We have revised the manuscript based on their comments. The revised sections are marked in red for your and reviewer’s references. The responses to all questions are given below:

1) Mw and purity of reagent were missed in experimental section.

Thank you for your careful checking. We have added the Mw and purity of all reagent in Section 2.1.

2) author should be explained the effects of nanorod shape on magnetic properties of products.

Thank you for your good suggestion. The ratio of length to diameter (L/D) has an obvious effect on the coercivity. And the increase of L/D can enhance the coercivity of Co nanorods, which is caused by the shape anisotropy. Generally, the Co nanorods with flat tips have smaller coercivity than cambered tips even though they have the similar L/D. Because Co nanorods with flat tips have more defects (stacking fault) between the tips and body, which can lead to a high demagnetizing filed and cause the low coercivity. (Adv. Nat. Sci.: Nanosci. Nanotechnol. 8 (2017) 025012).

The related descriptions have been added in our revised version.

3) it is important to use various surfactant agent in e self-assembly procedure.

Thank you for excellent suggestion. In fact, we also tried to use other surfactants to do the self-assembly. But they all failed. Thus, we did not highlight it in this work.

In our manuscript, the related descriptions can be found in experiment section. “The other surfactant like HDA and PVP, and other solution like hexane and chloroform were employed to replace the OA/OAm and methylbenzene, respectively. But they failed to get the well-aligned Co layers.”

4) JCPD card no is necessary for all XRD patterns.

Thank you for your carefully checking. We have added the JCPD card no (PDF#01-1278) in the revision.

5) Quality of Fig. 6 should be improved.

Thank you for your suggestion. A higher resolution picture has been instead of the old one, in this version.

6) Recent advances on synthesis, mechanism process and physicho-chemical properties are advised included, e.g. A: doi: 10.1016/j.actamat.2020.01.044, B: doi: 10.1039/D2NR00488G, C: doi: 10.1002/aenm.201502588, D: doi: 10.3390/nano12060982, E: doi: 10.1016/j.measurement.2021.110527, F: doi: 10.1016/j.sna.2021.112789, G: doi: 10.1016/j.measurement.2020.108409, H: doi: 10.1109/TTE.2021.3104876 and I: https://doi.org/10.1002/adem.202101680.

Thank you for raising this question. Some of them have been cited in the revision.

7) The novelty results should be highlighted in conclusion section.

Thank you for your good suggestion. We have added the related descriptions in conclusion section to highlight the novelty. In summary, we synthesized two kinds of Co nanorods, one with cambered tips and another with flat tips, and achieve a self-assembly of single layer of Co nanorods to study their magnetostatic interaction behavior. The as-synthesized Co nanorods with cambered tips have an average diameter of 10 nm and length of 100 nm with coercivity of 6.4 kOe, and flat-tip nanorods with a 30 nm diameter and 100 nm length exhibit a coercivity of 4.9 kOe.

We believe we have answered all questions raised  and revised the paper accordingly. The new version now is ready to be accepted for publication. Thank you for your time and consideration.

Best regards,

Zhenhui Ma

Reviewer 3 Report

The article consists of two unequal parts. If the reader may agree with the methods and approaches for the synthesis of cobalt nanoclusters, the interpretation of the results of magnetic measurements seems wrong. It is known that single-domain cobalt nanoclusters have a size of the order of 3-5 nm. This means that they are demagnetized at a temperature of the order of 14K ( https://www.researchgate.net/publication/11955368_Magnetic_Anisotropy_of_a_Single_Cobalt_Nanocluster

  ). This means that the coercive force for such objects must be very small.

Therefore, we can recommend to the authors two ways to improve the quality of the paper. The first is to abandon the interpretation of magnetization. I.e. it is necessary to leave in the article only methods of receiving and analyzing properties (microscopy, etc.).

The second way consists of a more detailed analysis of FC and ZFC dependences, as well as the function of nanocluster size distribution in order to find large cobalt particles, the presence of which can explain the magnetization curves at room temperature.

Author Response

Dear Reviewer, 

Thank you for your e-mail and forwarding the reviewers’ comments on our manuscript entitled “A self-assembly of single layer of Co nanorods to reveal the magnetostatic interaction mechanism”. We thank the reviewers for their constructive comments and kind recommendation. We have revised the manuscript based on their comments. The revised sections are marked in red for your and reviewer’s references. The responses to all questions are given below:

Thank you for your suggestions. The coercive mechanism of Co nanorods is totally different from small-sized Co nanoparticles or clusters. As we known, the magnetic particles have the largest coercivity at the key size from multiple-domain to single-domain structure, and the key size is about 30 nm for Co particles. (Chem. Rev., acs.chemrev.1c00860) Thus, our Co nanorods with a diameter about 10 nm and length 100-150 nm display a relatively higher coercivity than small-sized Co particles (several nanometers). Furthermore, these Co nanorods have been confirmed to be single-domain structure in previous work. (Scientific Reports 2014, 4.) And for Co clusters with size of 3-5 nm, they are superparamagnetic and thus have no coercivity at room temperature since their magnetocrystalline anisotropy at this size cannot compete with thermal disturbance. On the other hand, the coercivity of hcp-Co nanorods is mainly originated from shape anisotropy rather than magnetocrystalline anisotropy. In other words, the high L/D (ratio of length to diameter) lead to their large coercivity, which can be confirmed by many reports.

These descriptions have been in added section 3.2 in revised manuscript.

We believe we have answered all questions raised and revised the paper accordingly. The new version now is ready to be accepted for publication. Thank you for your time and consideration.

Best regards,

Zhenhui Ma

Reviewer 4 Report

The authors carried out interesting study and I recommend to publish the manuscript. I just recommend to try to perform some estimation and discussion of the quantitative parameters, which control the way of Co nanorods alignment. The discussion presented in section 3.3 is very qualitative, and, as a consequence, seems to be apparent to the reader. Speaking about quantitative parameters, two types of Co particles (Co nanorods with cambered tips and with flat tips) have ca 20 % difference in the coercivity (Hc = 6.4 and 4.9 kOe). Is it the single quantitative criterion which is responsible for different alignment of the nanorods?

Author Response

Dear Reviewer, 

Thank you for your e-mail and forwarding the reviewers’ comments on our manuscript entitled “A self-assembly of single layer of Co nanorods to reveal the magnetostatic interaction mechanism”. We thank the reviewers for their constructive comments and kind recommendation. We have revised the manuscript based on their comments. The revised sections are marked in red for your and reviewer’s references. The responses to all questions are given below:

Thank you for your good suggestions. The oriented degree does have an important influence on the coercivity. In our work, to avoid this effect, we measured the magnetic properties using the random samples other than aligned samples. These aligned samples (self-assembly samples) are only measured using XRD and TEM. Therefore, we think that the difference of their coercivity is mainly from their L/D (ratio of length to diameter) and the stacking fault in Co nanorods with flat tips, which is also confirmed by previous reports. (Acta Materialia 2018, 145, 290-297, Adv. Nat. Sci.: Nanosci. Nanotechnol. 8 (2017) 025012).

The measurement of magnetic property in detail was demonstrated in Section 2.6.

We believe we have answered all questions raised and revised the paper accordingly. The new version now is ready to be accepted for publication. Thank you for your time and consideration.

Best regards,

Zhenhui Ma

Round 2

Reviewer 3 Report

 The authors have done a lot of work to improve the article's quality in a short period of time.

The article therefore deserves to be published.